# Characteristics of the *AT-Hook Motif Containing Nuclear Localized* (*AHL*) Genes in Carrot Provides Insight into Their Role in Plant Growth and Storage Root Development

**DOI:** 10.3390/genes12050764

**Published:** 2021-05-18

**Authors:** Gabriela Machaj, Dariusz Grzebelus

**Affiliations:** Department of Plant Biology and Biotechnology, Faculty of Biotechnology and Horticulture, University of Agriculture in Krakow, 31-120 Krakow, Poland; gabriela.machaj@student.urk.edu.pl

**Keywords:** AT-hook motif, coexpression network analysis, *Daucus carota* L, development, *AHL* gene family, PPC domain, phylogeny, transcriptome

## Abstract

The *AT-hook motif containing nuclear localized (AHL)* gene family, controlling various developmental processes, is conserved in land plants. They comprise Plant and Prokaryote Conserved (PPC) domain and one or two AT-hook motifs. *DcAHLc1* has been proposed as a candidate gene governing the formation of the carrot storage root. We identified and in-silico characterized carrot AHL proteins, performed phylogenetic analyses, investigated their expression profiles and constructed gene coexpression networks. We found 47 *AHL* genes in carrot and grouped them into two clades, A and B, comprising 29 and 18 genes, respectively. Within Clade-A, we distinguished three subclades, one of them grouping noncanonical *AHLs* differing in their structure (two PPC domains) and/or cellular localization (not nucleus). Coexpression network analysis attributed *AHLs* expressed in carrot roots into four of the 72 clusters, some of them showing a large number of interactions. Determination of expression profiles of *AHL* genes in various tissues and samples provided basis to hypothesize on their possible roles in the development of the carrot storage root. We identified a group of rapidly evolving noncanonical *AHLs*, possibly differing functionally from typical *AHLs*, as suggested by their expression profiles and their predicted cellular localization. We pointed at several *AHLs* likely involved in the development of the carrot storage root.

## 1. Introduction

The *AT-hook motif containing nuclear localized* (*AHL*) gene family is common and conserved in all land plants, suggesting that *AHL* genes are important for plant growth and development. However, little is known about their precise roles. Functional analysis of some *AHLs* has been mostly limited to *Arabidopsis thaliana*. Some *AtAHL* genes have been described as controlling hypocotyl growth and senescence [1,2,3], fertility and pollen development [4,5], root development [6] and flowering [7,8]. Additionally, they may be involved in the regulation of primary metabolism [9], hormonal homeostasis and signaling [10,11], drought tolerance [11] and pathogen defense [12]. Beside *A. thaliana*, an *AHL* gene was reported to be required for the development of maize ears [13]; *depressed palea1 (dp1)* gene involved in floral development in rice was attributed to the *AHL* family [14] and members of the *AHL* gene family were reported as controlling pathogen defense response in chickpea [15].

On the structural level, AHL proteins comprise two conserved units, ca. 120-aa-long Plant and Prokaryote Conserved (PPC) domain (previously annotated as DUF296) and a region carrying one or two AT-hook motifs [3]. The AT-hook motif contains a core sequence Arg-Gly-Arg (flanked by Arg-Lys/Pro) capable of binding to AT-rich regions of B-form DNA [16]. AT-hook motifs have been divided into two types depending on the sequence variation at the C-terminus of the Arg-Gly-Arg core. Type-I contains Gly-Ser-Lys-Asn-Lys consensus sequence whereas Type-II carries Arg-Lys-Tyr [17]. The PPC domain is located at the carboxyl end relative to the AT-hook motif(s) [17]. The PPC domain is responsible for protein–protein interactions, facilitating the formation of trimers with other AHLs (forming homo-complexes), as well as with other nuclear proteins (e.g., histones and transcription factors), resulting in hetero-complexes. The conserved core of the PPC domain (Gly-Arg-Phe-Glu-Ile-Leu) seems to be essential for complex formation with transcription factors [3]. Two types of the PPC domain are distinguished depending on the amino acid sequence of the region upstream the conserved core motif. Type-A contains conserved Leu-Arg-Ser-His sequence whereas in Type-B Phe-Thr-Pro-His is the conserved sequence. Thus, based on the composition of AT-hook motif(s) and the type of the PPC domain, AHL proteins have been classified into three subtypes. Type-I AHLs contain one AT-hook motif of Type-I and Type-A PPC domain, Type-II AHLs contain two AT-hook motifs (Type-I and Type-II) and Type-B PPC domain, whereas Type-III AHLs contain one Type-II AT-hook motif and one Type-B PPC domain. Phylogenetic analysis of AHL gene families from 19 land plant species indicated that all Type-I AHLs belonged to Clade-A while Type-II and Type-III AHLs were grouped in Clade-B [18].

The cultivated carrot (*Daucus carota* subsp. *sativus*) is among the top 10 vegetable crops globally in terms of the area of production and market value. The worldwide carrot production grows steadily, possibly owing to increased awareness of health benefits associated with carrot consumption [19]. The carrot genome, comprising 473 Mb, has been recently sequenced [20] and a member of the *AHL* family, named *DcAHLc1*, has been proposed as a candidate gene governing the formation of the fleshy storage root [21]. However, the *AHL* gene family in carrot has not been thoroughly characterized. Here, we present global structural and functional characteristics of the *AHL* gene family in carrot, providing insight into the role of *AHLs* in developmental processes, including their possible involvement in the development storage roots.

## 2. Materials and Methods

### 2.1. Identification and In-Silico Characterization of AHL Proteins

Protein and transcript sequences from the carrot reference genome [20] were retrieved from NCBI (NCBI Annotation Release 100, 27 June 2016). InterProScan 5 v.5.38-76.0 [22] (with options: -appl Pfam; -dp; -iprlookup) was used to mine for carrot proteins carrying the PPC/DUF296 domain (Pfam ID PF03479; Available online: https://pfam.xfam.org/ (accessed on 7 October 2020)). The presence of the domain was further confirmed using the Conserved Domains Database (Available online: https://www.ncbi.nlm.nih.gov/Structure/cdd/wrpsb.cgi (accessed on 15 September 2020)). Motifs in protein sequences were predicted by MEME v.5.1.0 [23] with e-value 10^−8^ and parameters: -protein -oc. -nostatus -time 18,000 -mod anr -nmotifs 10 -minw 4 -maxw 160 -objfun classic -markov_order 0. Unique sequences containing both PPC domain(s) and AT-hook motif(s) were selected as representatives of the *AHL* gene family. Protein isoelectric points (pI) and molecular weight (MW) were predicted using Expasy Server (Available online: https://web.expasy.org/compute_pi/ (accessed on 16 September 2020)) whereas subcellular localization and signal peptides were predicted using CELO v.2.5 [24], (Available online: http://cello.life.nctu.edu.tw/ (accessed on 16 September 2020)) and SignalP 5.0 Server [25], (Available online: http://www.cbs.dtu.dk/services/SignalP/ (accessed on 16 September 2020)), respectively. Structures of the carrot *AHL* genes were determined by aligning respective coding sequences to the reference genome assembly [20] (NCBI accession LNRQ01000000).

### 2.2. Phylogenetic Analysis

To infer the evolution of carrot AHLs, we retrieved 29 *A. thaliana* AHL protein sequences from the TAIR database (Available online: https://www.arabidopsis.org/ (accessed on 21 September 2020) and the PPC domain of *Escherichia coli* from InterPro database (Available online: https://www.ebi.ac.uk/interpro/ (accessed on 21 September 2020)), and used them together with the carrot AHLs to construct a phylogenetic tree. All protein sequences (represented by the longest isoforms) were aligned in MUSCLE [26] using default parameters. MEGA-X [27] was used to find the best model and to construct a maximum likelihood (ML) tree based on 1000 bootstrap replicates. 

### 2.3. Gene Duplication Events

Carrot AHL proteins were used as queries for BLASTp against the carrot protein database at e ≤ 1 × 10^5^. To identify whole genome, segmental, tandem, proximal, and dispersed gene duplication events we used the top five hits for the duplicate_gene_classifier function in MCScanX [28] using default parameters.

### 2.4. Analysis of Gene Expression 

We analyzed differential expression patterns of the carrot *AHL* genes using previously reported transcriptomic data from the wild and the cultivated carrots [29]. To determine tissue-specific vs. constitutive gene expression patterns, we also used transcriptomic data from 20 different tissues of the reference carrot line DH1 [20] (PRJNA291977). Kallisto v.0.44.0 [30] was used to estimate the expression of *AHL* genes in each tissue separately. Read counts were standardized using median-by-ratio normalization implemented in EBSeq v.1.12.0 package [31] and then log + 1-transformed. 

### 2.5. Gene Coexpression Network Analysis

To develop coexpression networks, we used 47 transcriptomic data from the wild (9 samples) and the cultivated carrots (9 samples) [29] and 29 carrot root tissues [32] (BioProject PRJNA350691). Reads counts obtained from Kallisto v.0.44.0 [30] were normalized using the median normalization method in the EBSeq v.1.12.0 R package [31]. Correlations between carrot *AHL* genes and other genes differentially expressed during cultivated carrot development, as well as between the cultivated and the wild carrots (FDR < 0.05) [29], were calculated in Psych v.2.0.9 R package [33] using Pearson correlation and filtered by adjusted *p*-value < 0.01. Network statistics were calculated in Igraph v.1.2.6 R package [34]. Cytoscape v.3.7.2 [35] and clusterMaker v.1.3.1 plug-in [36] were used to construct coexpression networks of *AHL* genes and coexpressed genes. Functional annotation of all carrot genes was made using GO FEAD [37]. Analysis and visualization of Gene Ontology terms in the constructed networks was performed in Cytoscape with ClueGO v.2.5.7 [38] plug-in.

## 3. Results and Discussion

### 3.1. Identification and Characterization of Carrot AHLs

According to the annotation of the carrot reference genome (NCBI Annotation Release 100, 27 June 2016), there were 61 *AHL* genes. However, only 45 of them encoded proteins carrying both the PPC-domain and the AT-hook motif(s). Using a genome-wide scan, we also identified one additional gene (LOC108192665) that also encoded a protein comprising both AHL functional units. Additionally, in subsequent analyses we also included a gene (LOC108207572) which, according to results produced by InterProScan, encoded a protein with an incomplete PPC-domain and an AT-hook motif. Thus, we further characterized 47 putative carrot *AHL* genes (Table 1, Appendix A in the Appendix A).

Carrot AHL proteins carried one or two AT-hook motifs of Type-I or Type-II and one or two PPC domains (Figure 1). The conserved amino acid core of the PPC domain (Gly-Arg-Phe-Glu-Ile-Leu) was found in ca. 40% of the analyzed proteins, as compared to 62% in *A. thaliana* [3].

Phylogenetic analysis of AHLs from carrot and *A. thaliana* showed diversification of carrot AHLs into two major clades, as described previously in other species [18,19,39], Clade-A and Clade-B, comprising 29 and 18 carrot AHLs, respectively (Figure 2).

Within Clade-A three subclades, A1, A2, and A3, could be distinguished. The canonical Type-I AHLs were grouped in subclade A1, containing 17 carrot proteins showing clear relationships to their homologs from *A. thaliana*. All of them contained one Type-I AT-hook motif Arg-Gly-Arg-Pro, followed by Gly-Ser-Lys-Asn-Lys-Pro-Lys on the C-terminus and Type-A PPC domain starting with Leu-Arg-Ser-His on the *n*-terminus (Figure 1). The remaining 12 proteins assigned to subclades A2 and A3 had shorter and less conserved PCC domains (Figure 1) and showed different characteristics. In subclade A2, carrot DcAHL19 and DcAHL33 clustered with AtAHL17 and AtAHL28, while carrot proteins in subclade A3, carrying one AT-hook motif and one or two PPC domains, were not associated with any AHL from *A. thaliana* (Figure 2).

In Clade-B, AHLs carried Type-B PPC domain with a highly conserved sequence Phe-Thr-Pro-His on the *n*-terminus and one or two AT-hook motifs (Figure 1). Depending on the AT-hook motif composition, proteins assigned to Clade-B could be divided into Type-II and Type-III [3]. In Type-II, most carrot AHLs proteins (61%) contained two types of the AT-hook motif, i.e., Type-I with a conserved core Arg-Gly-Arg-Pro followed by less conserved amino acids (Figure 1 and Figure 3B), and Type-II with a highly conserved Arg-Gly-Arg-Pro-Arg-Lys-Tyr core (Figure 1 and Figure 3C). Generally, Type-I AT-hook motifs in Clade-B were less conserved than those in Clade-A (Figure 3A,B). Three proteins carrying only one Type-I AT-hook motif also clustered with Type-II AHLs (Figure 1). A similar observation has been reported previously for two maize AHLs [40]. Type-III AHLs contained only one AT-hook motif of Type-II with a conserved Arg-Gly-Arg-Pro-Arg-Lys-Tyr core (Figure 1 and Figure 3B). As in other plant species [18,19,39,40], Type-II and Type-III carrot AHLs did not form distinct subclades within Clade-B (Figure 2).

The lack of separation of Type-II/III AHLs and the presence of proteins with one AT-hook motif of Type-I within Type-II AHLs, jointly support relatively recent divergence of these two protein types and multiple transitions between Type-II and Type-III. Phylogeny of *A. thaliana* and carrot AHLs support the hypothesis that Clade-B evolved from Clade-A [3,22,23].

### 3.2. Structure of AHL Genes

As expected, genes grouped in subclades A1 and A2 (canonical Clade-A *AHLs*) were intronless, whereas in Clade-B Type-II *AHLs* contained five exons and in Type-III *AHLs* the number of exons ranged from five (three genes) to six (one gene) (Figure 1). Positions of exon/intron junctions in Clade-B genes were conserved. Within subclade A3, six genes, *DcAHL7-DcAHL10*, *DcAHL35* and *DcAHL36*, were intronless, while the remaining four, *DcAHL43*, *DcAHL44*, *DcAHL45*, and *DcAHL47*, showed an unusual structure, as they all harbored two introns, with exceptionally large intron I, and carried two PPC domains (Figure 1).

### 3.3. Duplication Events

Gene families in plants may enlarge either as a result of whole genome duplication (WGD) events or by segmental duplication events including tandem, proximal and dispersed duplications [41]. In carrot, two lineage-specific WGDs have contributed to the expansion of carrot gene families [20]. Similar to maize, *A. thaliana* and rice [40,42], mostly dispersed duplications contributed the divergence of the *AHL* gene family in carrot. Thirty-seven (ca. 80%) carrot *AHLs* evolved through dispersed duplications. Besides dispersed duplication events, we also observed four tandem and five proximal gene duplications contributing to the *AHL* gene expansion in carrot (Figure 1). Interestingly, most of them were observed within subclade A3, possibly pointing at a recent dynamic evolution of those noncanonical *AHLs* in carrot. Proximal duplication events were revealed for two groups, i.e., *DcAHL7* and *DcAHL10* located on chromosome 2, and *DcAHL43*, *DcAHL44* and *DcAHL45* located on chromosome 9. The latter three genes sharing the unusual structure (two introns and two PPC domains) were expressed in germinating seeds (Figure 4). *DcAHL10* had high or moderate expression in most tissues while its proximal copy *DcAHL7* showed no expression. Possibly, expression of *DcAHL7* may be triggered environmentally, e.g., by an abiotic or biotic stress, or the gene could be nonfunctional. Tandem duplication events were detected for *DcAHL8* and *DcAHL9*, located on chromosome 2. *DcAHL8* was weakly expressed in flowers and seeds, whereas its counterpart *DcAHL9* exhibited moderate expression levels in most tissues (Figure 4), possibly pointing at their sub- or neo-functionalization, supported also by different predicted cellular localizations of DcAHL8 and DcAHL9 proteins, as described below.

### 3.4. Properties of the Predicted Proteins

Based on in-silico predictions, carrot AHL proteins were on average 300 aa long, ranging from 157 aa (XP_017233492.1) to 454 aa (XP_017243692.1). The predicted average molecular weight of carrot AHL proteins was 31.11 kDa with an isoelectric point (pI) between 4.41 and 10.24 (Appendix A in the Appendix A). Thirty-seven *DcAHLs* produced proteins which were predicted to be localized in the nucleus, (consistent with their ability to bind AT-rich regions of DNA [16]). However, some proteins encoded by *DcAHLs*, mostly those grouped in subclade A3, were predicted to be targeted towards other cellular components, e.g., the plasma membrane (five proteins, including the four proteins from subclade A3 carrying two PPC domains, i.e., DcAHL43, DcAHL44, DcAHL45, and DcAHL47), chloroplasts (four proteins, including the three proteins from subclade A3, i.e., DcAHL7, DcAHL8 and DcAHL10) and extracellular space (DcAHL9 from subclade A3) (Appendix A in the Appendix A). The plastid localization of the four DcAHL proteins may suggest they interact with cpDNA [40], while the predicted extracellular localization of DcAHL9, carrying a signal peptide (Sec/SPI) (*p* = 0.4727) with a cleavage site ASG-LP between positions 29 and 30 (Appendix A in the Appendix A), is surprising.

### 3.5. Expression of AHL Genes

In order to examine expression patterns of *AHL* genes in carrot, we collated transcriptomic data from 20 DH samples derived from various tissues and organs of carrot (data from [20]) with those obtained for wild and cultivated carrot seedlings, developing roots and mature roots [29].

Three of the 47 carrots *AHL* genes (*DcAHL7*, *DcAHL28*, *DcAHL35*) were not expressed in any of the analyzed samples (Figure 4). Nineteen carrot *AHL* genes (40%) exhibited constitutive, high or moderate levels of expression across all examined samples (Figure 4). Fourteen of the 19 constitutively expressed *AHLs* belonged to Clade-B. In that clade, only four genes, i.e., *DcAHL6*, *DcAHL21*, *DcAHL24*, and *DcAHL29*, were not constitutively expressed. In contrast, in Clade-A only five genes *DcAHL19*, *DcAHL23*, *DcAHL31*, *DcAHL32*, and *DcAHL34* were constitutively expressed, while the remaining genes showed tissue-specific expression.

Expression-based clustering of carrot *AHLs* revealed two clusters, one grouping constitutively expressed *AHLs* and the other grouping two constitutively expressed and all tissue-specific *AHLs* (Figure 4). Within the second cluster one can distinguish a subcluster comprising eight genes (*DcAHL12*, *DcAHL15*, *DcAHL18*, *DcAHL41*, *DcAHL37*, *DcAHL39*, *DcAHL42*, and *DcAHL46*) expressed primarily in roots. In particular, *DcAHL37*, *DcAHL41*, and *DcAHL46* were highly expressed in carrot roots, while they showed little or no expression in other organs.

Hierarchical clustering based on the expression of *AHL* genes in different organs revealed three clusters. Cluster I grouped samples from leaves, petioles, unopened flowers and bracts from open flowers. Cluster 2 grouped samples from seedlings and roots at different stages of development. In that cluster, a group of samples comprising developing cultivated roots (C2), phloem, xylem and hypocotyls formed a subcluster, while samples representing wild roots, mature storage roots and seedlings formed a separate subcluster. Cluster 3 grouped samples from opened flowers, buds, bracts from unopened flowers, fibrous roots, callus and germinating seeds.

The clustering results indicate that *AHL* genes are developmentally regulated. Nineteen of those were expressed in all examined tissues/organs, albeit at different levels, while 25 exhibited tissue/organ-specific expression varying in the course of development. These results provide insight into possible biological roles of specific *AHLs* in the development of carrot organs and constitute the basis for further investigation.

### 3.6. Coexpression Networks of AHL Genes Expressed in the Storage Root

The coexpression analysis was performed on 7794 genes identified as differentially expressed in the roots of cultivated and wild carrots, as well as during storage root development in cultivated carrots, as reported previously by Machaj et al. (2018) [29], including 35 *DcAHL* genes expressed in carrot roots. In the final network, 15 *DcAHLs* were not considered due to low correlation and high corresponding *p*-values. The coexpression analysis resulted in the construction of an extensive network containing 6798 nodes grouped into 72 clusters (Appendix A, Appendix A in the Appendix A). Correlation values in the network ranged from −0.968 to −0.766 for negatively correlated genes and from 0.766 to 1 for positively correlated genes. *DcAHLs* in the network were assigned to four clusters (Figure 5; yellow nodes).

Cluster-1 (Appendix A (green) in the Appendix A) comprised 3743 genes including nine *AHLs* (Table 2; Figure 5A) (clustering coefficient 0.519). Gene ontology (GO) analysis for Cluster-1 indicated photosynthesis, pigment biosynthetic process, nucleic acid metabolic process, cellular component/cell wall biogenesis, regulation of RNA biosynthetic process, carbohydrate metabolic process and cell division as the most enriched biological terms (Appendix A in the Appendix A). *DcAHLs* from Cluster-1 showed 229 direct interactions (Figure 5A, Appendix A in the Appendix A). While 218 interactions were within Clusters-1, 8 and 3 interactions with genes from Cluster-2 and Cluster-13 were observed, respectively. *DcAHL16* was coexpressed with 62 genes. GO analysis of genes coexpressed with *DcAHL16* indicated its involvement in fucose/hexose, amino sugar and nucleotide sugar metabolism; steroid biosynthesis and protein import to nucleus (Figure 6A). The high number of connections for *DcAHL16* indicated that it was a prominent regulatory gene in Cluster-1. Interestingly, this gene (reported as *DcAHLc1* [21]) was previously identified as a candidate domestication gene and shown to be differentially expressed during root development in cultivated and wild carrots [29]. *DcAHL11*, interacting with 51 genes, is another important gene in Cluster-1. GO analysis of the first neighbors of *DcAHL11* implied that the gene was likely involved in the regulation of cell division, cellulose metabolism and protein phosphorylation (Figure 6B).

Cluster-2 (Appendix A (red) in the Appendix A) comprised 2314 genes including eight *AHLs* (clustering coefficient 0.615; Table 2; Figure 5B). *AHLs* from Cluster-2 demonstrated 2416 direct interactions with other genes, including 2312 interactions within Cluster-2 followed by 79, 17, and 3 interactions with genes from Cluster-1, Cluster-3 and Cluster-13, respectively (Appendix A in the Appendix A). GO enrichment analysis for Cluster-2 indicated that the most significantly enriched biological processes were associated with transport (ion, nitrogen compound, protein, endosomal and vesicle-mediated transport), plastid organization and tRNA/mRNA/rRNA metabolic process. A group of terms associated with root development, multicellular organism development, plant organ morphogenesis, mitotic cell cycle, cell/organ differentiation/morphogenesis and development; cell wall biogenesis and starch/polysaccharide metabolism could also be distinguished (Appendix A in the Appendix A). *DcAHL38* showed 825 direct interactions in Cluster-2. GO enrichment analysis indicated that it might be involved in a range of biological processes, e.g., intracellular protein transport; regulation of immune system and meristem structural organization, as well as anatomical structure morphogenesis and calcium-mediated signaling (Figure 6C). Interestingly, *DcAHL38* showed the strongest correlation (0.934; *p* = 3.431 × 10^14^) with *BYPASS1* (LOC108196316). In *A. thaliana*, *BYPASS1* is required for normal root and shoot development by preventing constitutive production of a root mobile carotenoid-derived signaling compound that is capable of arresting shoot and leaf development [43,44]. Moreover, *DcAHL38* was coexpressed with genes LOC108214645 (0.924, *p* = 6.714 × 10^13^) and LOC108218757 which in *A. thaliana* are involved in the regulation of root stem cell niche identity [45] and responsible for normal plant growth and development [46]. Thus, *DcAHL38* is possibly involved in the regulation of storage root morphogenesis in carrot. Additionally, *DcAHL38* shows a positive correlation with four other *AHL* genes (*DcAHL13*, *DcAHL26*, *DcAHL27*, and *DcAHL30*) from Cluster-2, forming a group of five interrelated *DcAHLs* (Figure 5F). As previously reported, AHLs regulate plant growth and development by forming homo- or hetero-trimeric complexes via their PPC domains [3]. Our results suggest that the potential direct interactions among the five DcAHL proteins might be important for the regulation of storage root development, however, experimental validation of those interactions is required to support the hypothesis.

Cluster-3 (Appendix A (blue) in the Appendix A) comprised 289 genes including two *AHLs* (clustering coefficient 0.253; Table 2; Figure 5C). In that cluster, *DcAHL19* interacted with 17 genes (four from Cluster-3, six from Cluster-1 and seven from Cluster-2), while *DcAHL42* interacted (negative correlation) with just one gene. Notably, the expression of *DcAHL19* was reported as highly varying in the course of the storage root development [29]. GO analysis of Cluster-3 indicated that it was primarily associated with RNA splicing, cellular response to oxidative stress, fruit/seed development and xyloglucan metabolic process (Appendix A in the Appendix A).

Cluster-4 (Appendix A (pink) in the Appendix A) comprised nine genes including *DcAHL37* (clustering coefficient 0.504; Table 2; Figure 5D). In that cluster, all genes were positively correlated. *DcAHL37* interacted with 12 genes, only five of them being assigned to Cluster-4. Due to the small number of genes in that cluster, it was not possible to perform GO enrichment. Nevertheless, *DcAHL37* had very high ‘betweenness centrality’ (b.c. = 14,455.308) and was highly expressed in roots (as compared to other carrot organs), which indicates that it might be important in process of the carrot storage root development.

We conclude that *DcAHL* genes assigned to the above described clusters are likely candidates governing storage root development in carrot. To more precisely investigate their role, we performed GO enrichment analysis of all first neighbors to *DcAHLs* (direct interactions) and compared it with GO terms for genes assigned to clusters 5-72. For genes not coexpressed with *DcAHLs*, biological processes associated with macromolecule metabolism, reactive oxygen species biosynthesis, oxidation–reduction, nitrogen metabolism, methylation and post-transcriptional gene silencing were highly enriched (Appendix A in the Appendix A). Genes coexpressed with *DcAHLs* were mainly associated with cellular macromolecule localization, small molecule metabolism, anatomical structure development/arrangement, cell wall organization or biogenesis, biosynthesis, pigment metabolism, cell communication and division (Figure 7), suggesting that they are important determinants of the storage root development in cultivated carrots.

## 4. Conclusions

Using a genome-wide scan and transcriptomic evidence, we identified and characterized the *AHL* gene family in carrot. We identified a group of noncanonical *AHLs* (subclade A3) which likely evolved rapidly and might have acquired functions different from typical *AHLs*, as suggested by their expression profiles and their predicted cellular localization. Gene coexpression network analysis provided evidence for the involvement of several *AHLs* in the development of the storage root in carrot.

## Figures and Tables

**Figure 1 genes-12-00764-f001:**
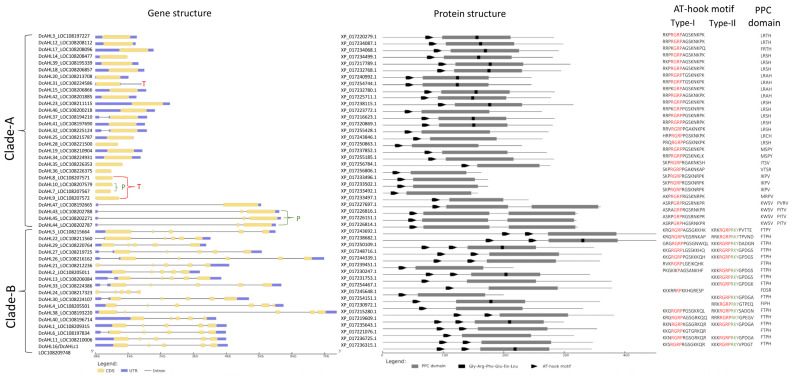
Structural characteristics of carrot *AHL* genes and proteins; left panel: exons are shown as yellow boxes, 5′- and 3′UTRs are shown as blue boxes, introns are shown as black lines, tandem and proximal duplication events are marked with T and P, respectively; right panel; PPC domain are shown as gray boxes, positions of the Gly-Arg-Phe-Glu-Ile-Leu core motifs within PPC domains are shown as black boxes, AT-hook motifs are shown as black arrowheads. Amino acid sequences of AT-hook motifs and the *n*-termini of the PPC domains are shown.

**Figure 2 genes-12-00764-f002:**
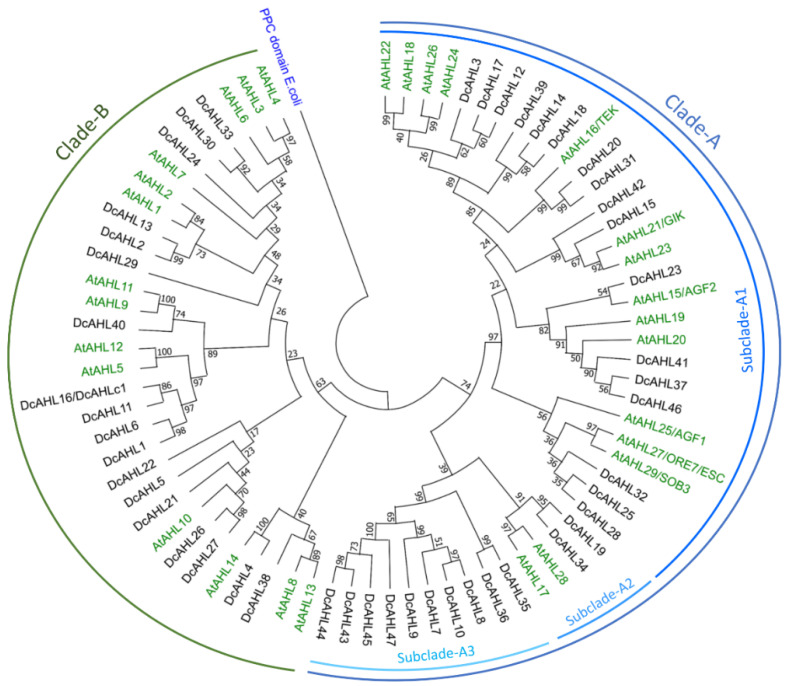
Maximum likelihood tree (1000 bootstrap replicates) of AHL proteins in carrot (black) and *A. thaliana* (green); PPC domain from *E. coli* (blue) was used to root the tree.

**Figure 3 genes-12-00764-f003:**
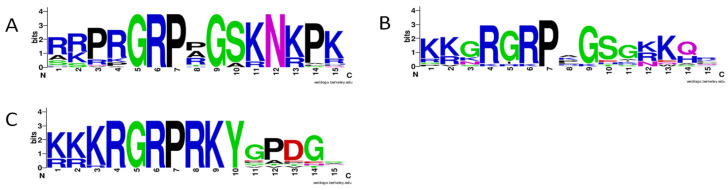
AT-hook motif types present in carrot AHL proteins: Type-I in Clade-A (**A**); Type-I in Clade-B (**B**); Type-II (**C**).

**Figure 4 genes-12-00764-f004:**
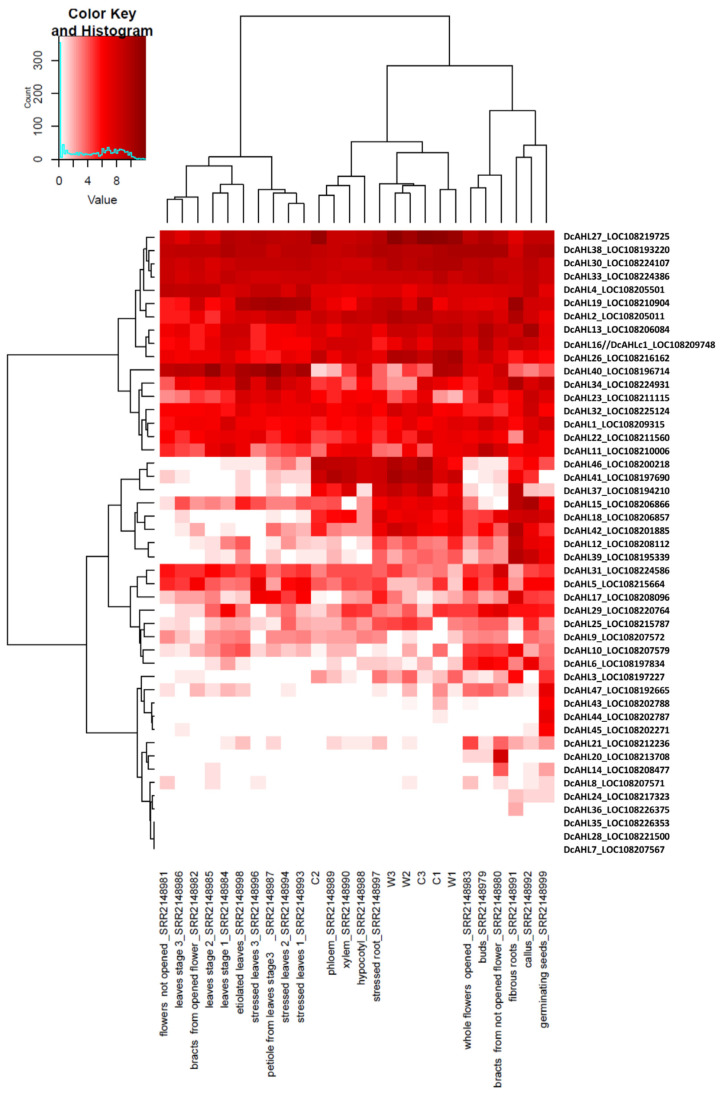
Hierarchical clustering and expression patterns of *AHL* genes in seedlings (C1, W1), developing (C2, W2) and mature (C3, W3) roots from cultivated (C) and wild carrot (W) (transcriptome data from [29]) and 20 tissues of the carrot reference line DH1 (germinating seeds, hypocotyl, leaves stage 1, 2, and 3, petioles from leaves at stage 3, stressed leaves stage 1, 2, and 3, etiolated leaves, fibrous root, root xylem, root phloem, stressed root, buds, flowers not opened, whole flowers opened, bracts from not opened flowers, bracts from opened flowers, and callus; transcriptome data from [20]). The color scale represents log2(normalized_reads + 1) values.

**Figure 5 genes-12-00764-f005:**
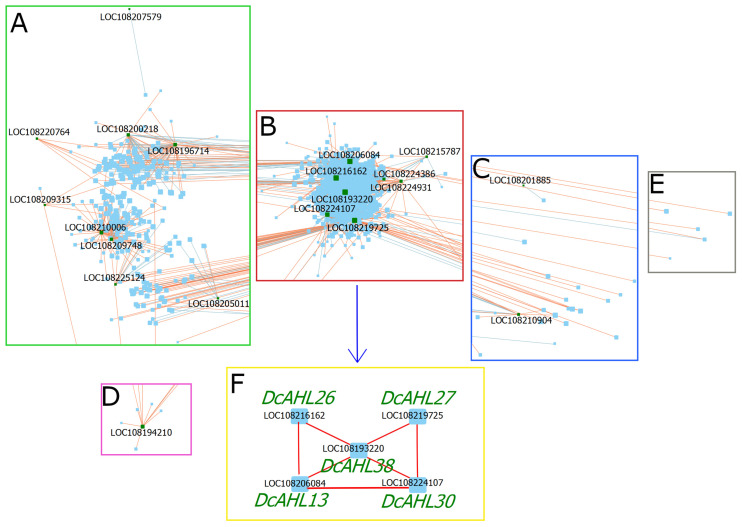
The coexpression network of genes differentially expressed in carrot storage roots. (**A**–**E**) Five clusters comprising *DcAHL* genes (shown as green nodes) and/or their first neighbors (shown as blue nodes); red and blue lines connecting genes represent positive and negative correlations between those genes, respectively. (**F**) A group of five potentially directly interacting *DcAHL* genes in Cluster-2.

**Figure 6 genes-12-00764-f006:**
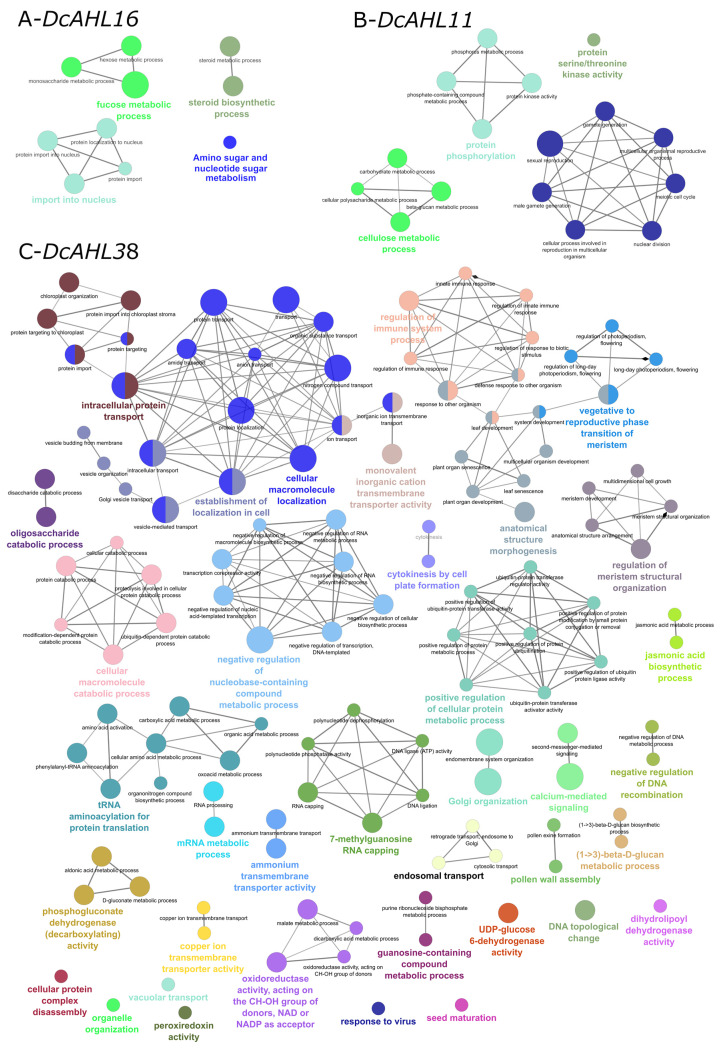
GO enrichment results for carrot genes possibly interacting with *DcAHL16* (**A**); *DcAHL11* (**B**); *DcAHL38* (**C**). Sizes of circles show the number of genes corresponding to GO terms.

**Figure 7 genes-12-00764-f007:**
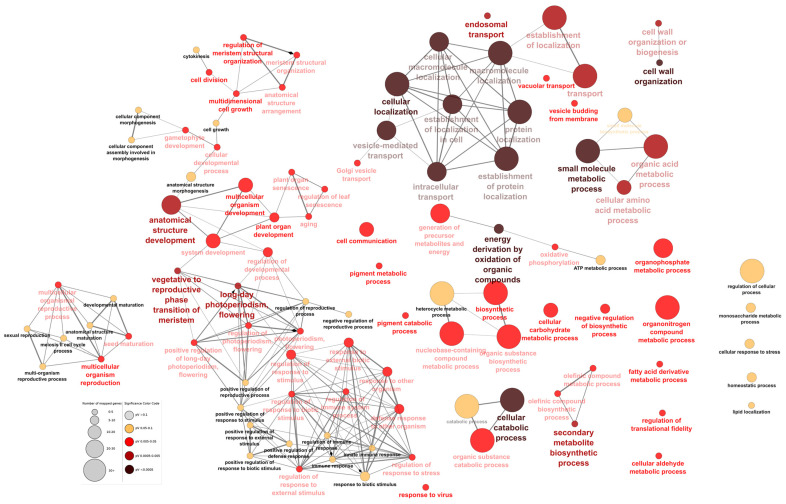
GO enrichment results for carrot genes possibly interacting with *DcAHLs*. Sizes of circles show the number of genes corresponding to GO terms, colors show significance of the enrichment, as shown in the significance legend box.

**Table 1 genes-12-00764-t001:** List of *AHL* genes identified in carrot.

Gene Name	Gene ID	Protein ID *	Chromosome	Gene Name	Gene ID	Protein ID *	Chromosome
*DcAHL1*	LOC108209315	XP_017235643.1	chr1	*DcAHL24*	LOC108217323	XP_017245648.1	chr4
*DcAHL2*	LOC108205011	XP_017230247.1	chr1	*DcAHL25*	LOC108215787	XP_017243846.1	chr4
*DcAHL3*	LOC108197227	XP_017220279.1	chr1	*DcAHL26*	LOC108216162	XP_017244339.1	chr4
*DcAHL4*	LOC108205501	XP_017230972.1	chr1	*DcAHL27*	LOC108219725	XP_017248716.1	chr5
*DcAHL5*	LOC108215664	XP_017243692.1	chr1	*DcAHL28*	LOC108221500	XP_017250863.1	chr5
*DcAHL6*	LOC108197834	XP_017221076.1	chr1	*DcAHL29*	LOC108220764	XP_017250109.1	chr5
*DcAHL7*	LOC108207567	XP_017233492.1	chr2	*DcAHL30*	LOC108224107	XP_017254151.1	chr5
*DcAHL8*	LOC108207571	XP_017233496.1	chr2	*DcAHL31*	LOC108224586	XP_017254744.1	chr6
*DcAHL9*	LOC108207572	XP_017233497.1	chr2	*DcAHL32*	LOC108225124	XP_017255428.1	chr6
*DcAHL10*	LOC108207579	XP_017233502.1	chr2	*DcAHL33*	LOC108224386	XP_017254467.1	chr6
*DcAHL11*	LOC108210006	XP_017236725.1	chr2	*DcAHL34*	LOC108224931	XP_017255185.1	chr6
*DcAHL12*	LOC108208112	XP_017234087.1	chr2	*DcAHL35*	LOC108226353	XP_017256784.1	chr6
*DcAHL13*	LOC108206084	XP_017231753.1	chr2	*DcAHL36*	LOC108226375	XP_017256806.1	chr6
*DcAHL14*	LOC108208477	XP_017234499.1	chr2	*DcAHL37*	LOC108194210	XP_017216623.1	chr7
*DcAHL15*	LOC108206866	XP_017232780.1	chr2	*DcAHL38*	LOC108193220	XP_017215280.1	chr7
*DcAHL16/*	LOC108209748	XP_017236315.1	chr2	*DcAHL39*	LOC108195339	XP_017217789.1	chr7
*DcAHLc1*				*DcAHL40*	LOC108196714	XP_017219609.1	chr7
*DcAHL17*	LOC108208096	XP_017234068.1	chr2	*DcAHL41*	LOC108197690	XP_017220869.1	chr8
*DcAHL18*	LOC108206857	XP_017232768.1	chr2	*DcAHL42*	LOC108201885	XP_017225711.1	chr9
*DcAHL19*	LOC108210904	XP_017237852.1	chr3	*DcAHL43*	LOC108202788	XP_017226816.1	chr9
*DcAHL20*	LOC108213708	XP_017240992.1	chr3	*DcAHL44*	LOC108202787	XP_017226814.1	chr9
*DcAHL21*	LOC108212236	XP_017239451.1	chr3	*DcAHL45*	LOC108202271	XP_017226151.1	chr9
*DcAHL22*	LOC108211560	XP_017238682.1	chr3	*DcAHL46*	LOC108200218	XP_017223772.1	chr9
*DcAHL23*	LOC108211115	XP_017238115.1	chr3	*DcAHL47*	LOC108192665	XP_017227697.1	chrUn_LNRQ01000029

* the largest isoform.

**Table 2 genes-12-00764-t002:** A list of *AHL* genes showing significant interactions in the coexpression network analysis of genes differentially expressed in the carrot storage roots.

Gene Name	GeneID	Cluster	Betweenness	Degree	Rank_stat
*DcAHL16//DcAHLc1*	LOC108209748	1	632.31	62	2695.25
*DcAHL11*	LOC108210006	1	757.12	51	2626.50
*DcAHL40*	LOC108196714	1	3490.89	46	3168.00
*DcAHL46*	LOC108200218	1	1361.78	41	2700.25
*DcAHL32*	LOC108225124	1	5.90	12	1268.00
*DcAHL2*	LOC108205011	1	1.70	6	951.75
*DcAHL29*	LOC108220764	1	9.82	6	1076.25
*DcAHL1*	LOC108209315	1	0.66	4	788.75
*DcAHL10*	LOC108207579	1	0.00	1	237.00
*DcAHL38*	LOC108193220	2	16,399.50	825	6140.50
*DcAHL26*	LOC108216162	2	6491.57	566	5180.00
*DcAHL13*	LOC108206084	2	10,024.21	372	5299.75
*DcAHL27*	LOC108219725	2	63,411.36	271	5842.75
*DcAHL30*	LOC108224107	2	6394.09	263	4611.25
*DcAHL33*	LOC108224386	2	22.44	81	2234.00
*DcAHL34*	LOC108224931	2	961.10	31	2474.00
*DcAHL25*	LOC108215787	2	48.34	7	1299.25
*DcAHL19*	LOC108210904	3	253.15	17	1894.25
*DcAHL42*	LOC108201885	3	0.00	1	237.00
*DcAHL37*	LOC108194210	4	14,455.31	12	3685.00

## Data Availability

Publicly available datasets were analyzed in this study. Source files can be downloaded from the GenBank Short Read Archive acc. no. SRP155333, PRJNA291977 and PRJNA350691.

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
