# Peer review of "Characteristics of the AT-Hook Motif Containing Nuclear Localized (AHL) Genes in Carrot Provides Insight into Their Role in Plant Growth and Storage Root Development"

_genes, 2021, doi:10.3390/genes12050764_

Round 1
Reviewer 1 Report
The minor errors are given below.
The abstract can be modified by avoiding subsections.
Line 53 & 54 - Based on the amino acid sequence of the region upstream the PPC domain, two types are distinguished. – needs modification
In Table 1 chromosome numbers can be included
i.e., and in-silico can be italicized
The quality of Figures 5 & 6 can be improved.
Generally, this is a good Bioinformatics paper. Can be a resource paper for undertaking further wet-lab experiments. Standard databases and methodologies are used in the investigation.
Author Response
Thank you for the comments.
The abstract can be modified by avoiding subsections.
A: Corrected, subsections were removed.
Line 53 & 54 - Based on the amino acid sequence of the region upstream the PPC domain, two types are distinguished. – needs modification
A: The sentence was modified: Two types of the PPC domain are distinguished depending on the amino acid sequence of the region upstream the conserved core motif.
In Table 1 chromosome numbers can be included
A: Done
i.e., and in-silico can be italicized
A: Done
The quality of Figures 5 & 6 can be improved.
A: In the submitted file, the figures were of good quality. From my experience as a reviewer and editor I know that in the working pdf file provided to reviewers quality of figures can sometimes be worsened. We will provide the source figures to the editor, if needed.
Generally, this is a good Bioinformatics paper. Can be a resource paper for undertaking further wet-lab experiments. Standard databases and methodologies are used in the investigation.
A: Thank you for your effort to review the manuscript.
Reviewer 2 Report
In this MS, authors identified 47 AHL genes in carrots, and analyzed their structure, expression pattern, and coexpression. However, some issue authors need to clarify or correct before publishing.
The methods authors used in the MS were not specifically described, authors only showed the packages used for specific analysis, however, most of the packages were under development and included different options, authors need to provide the detailed information, such as version, functions, options, so other people can replicate the results.
Authors constructed a co-expression network using transcriptomic data. Authors need to provide the detailed information, such as how authors process the sequencing, alignment. And how many transcriptomic data were used?
Authors mentioned several times "differentially expressed", however, how do authors quantify the differentially expressed gene? This needs detailed information.
Authors used the protein sequence to analyze the gene duplication events using the blastp package. Where did authors get the protein sequence data?
For results description, authors may want to use specific numbers, while not words like "most", "some". For example in "Properties of the predicted proteins" part, authors use "
Most AHLs were predicted to be localized in the nucleus (37 proteins)", why not directly point out the number of AHLs? and what does the 37 proteins mean?
In "3.5 Expression of AHL genes", authors wrote "Such clear clustering results, both tissue-wise and gene-wise", please clarify what is tissue-wise and gene-wide. Even though those genes are expressed in different tissues, how can authors conclude those genes involved plant development and organogenesis? I believe There is a gap between tissue specific expression and plant development and organogenesis.
In "3.6. Co-expression networks of AHL genes expressed in the storage root", authors filtered the network by "In the final network, 15 DcAHLs were not considered due to low correlation/p-values.", authors may want to say due to low correlation or high p-values?
Author Response
In this MS, authors identified 47 AHL genes in carrots, and analyzed their structure, expression pattern, and coexpression. However, some issue authors need to clarify or correct before publishing.
A: Thank you for providing insightful comments on our manuscript.
The methods authors used in the MS were not specifically described, authors only showed the packages used for specific analysis, however, most of the packages were under development and included different options, authors need to provide the detailed information, such as version, functions, options, so other people can replicate the results.
A: We added packages versions as requested. Options and functions were added only if they differ from default parameters.
Authors constructed a co-expression network using transcriptomic data. Authors need to provide the detailed information, such as how authors process the sequencing, alignment. And how many transcriptomic data were used?
A: We used transcriptomic data from 47 samples. Transcriptomic data were retrieved from NCBI so we did not perform any additional processing except Kallisto used to estimate read counts for genes, as specified in par. 2.4 (M&M).
Authors mentioned several times "differentially expressed", however, how do authors quantify the differentially expressed gene? This needs detailed information.
A: We used differentially expressed genes in the wild and the cultivated carrots, as reported previously (Machaj, G.; Bostan, H.; Macko-Podgórni, A.; Iorizzo, M.; Grzebelus, D. Comparative Transcriptomics of Root Development in Wild and Cultivated Carrots. Genes, 2018, 9, 431, doi:10.3390/genes9090431 – ref. [29]) where DEGs were called with FDR of 0.05 or less.
Authors used the protein sequence to analyze the gene duplication events using the blastp package. Where did authors get the protein sequence data?
A: Protein and transcript sequences from the carrot reference genome were retrieved from NCBI - an introductory sentence was added to par. 2.1 (M&M). Unique sequences containing both PPC domain(s) and AT-hook motif(s) were selected as representatives of the AHL gene family and were used for duplication analysis.
For results description, authors may want to use specific numbers, while not words like "most", "some". For example in "Properties of the predicted proteins" part, authors use " Most AHLs were predicted to be localized in the nucleus (37 proteins)", why not directly point out the number of AHLs? and what does the 37 proteins mean?
A: Multiple instances of imprecise gene/protein counts were corrected in the results. We also corrected the wording in the section indicated by the Reviewer.
In "3.5 Expression of AHL genes", authors wrote "Such clear clustering results, both tissue-wise and gene-wise", please clarify what is tissue-wise and gene-wide. Even though those genes are expressed in different tissues, how can authors conclude those genes involved plant development and organogenesis? I believe There is a gap between tissue specific expression and plant development and organogenesis.
A: We agree that the statement was too strong and replaced it with a more neutral: “The clustering results indicate that AHL genes are developmentally regulated.”
In "3.6. Co-expression networks of AHL genes expressed in the storage root", authors filtered the network by "In the final network, 15 DcAHLs were not considered due to low correlation/p-values.", authors may want to say due to low correlation or high p-values?
A: Thank you for pointing it out, we corrected it to: “…15 DcAHLs were not considered due to low correlation and high corresponding p-values.”
Round 2
Reviewer 2 Report
I don' have any comments at this time.
Author Response
We value the opinion of Reviewer 2 suggesting that all sections of the manuscript could still be improved. However, as no specific instructions aiming at further improvement have been given, we have not introduced any changes to the manuscript.